# *Pseudomonas* sp. COW3 Produces New Bananamide-Type Cyclic Lipopeptides with Antimicrobial Activity against *Pythium myriotylum* and *Pyricularia oryzae*

**DOI:** 10.3390/molecules24224170

**Published:** 2019-11-17

**Authors:** Olumide Owolabi Omoboye, Niels Geudens, Matthieu Duban, Mickaël Chevalier, Christophe Flahaut, José C. Martins, Valérie Leclère, Feyisara Eyiwumi Oni, Monica Höfte

**Affiliations:** 1Laboratory of Phytopathology, Department of Plants and Crops, Faculty of Bioscience Engineering, Ghent University, Coupure Links 653, B-9000 Ghent, Belgium; olumideowolabi.omoboye@ugent.be (O.O.O.); FeyisaraEyiwumi.Olorunleke@UGent.be (F.E.O.); 2Department of Microbiology, Obafemi Awolowo University, Ile-Ife 220005, Osun State, Nigeria; 3NMR and Structure Analysis Unit, Department of Organic and Macromolecular Chemistry, Faculty of Science, Ghent University, Krijgslaan 281, B-9000 Gent, Belgium; niels.geudens@ugent.be (N.G.); jose.martins@ugent.be (J.C.M.); 4Univ. Lille, INRA, ISA, Univ. Artois, Univ. Littoral Côte d’Opale, EA 7394-ICV- Institut Charles Viollette, F-59000 Lille, France; matthieu.duban@univ-lille.fr (M.D.); mickael.chevalier1@polytech-lille.fr (M.C.); christophe.flahaut@univ-artois.fr (C.F.); valerie.leclere@univ-lille.fr (V.L.)

**Keywords:** non-ribosomal peptide synthetases (NRPS), Kendrick mass defect, in silico analysis, hyphal leakage, extensive branching, vacuolization, mycophagy, *Pseudomonas koreensis* group, NMR spectroscopy, *Magnaporthe oryzae*

## Abstract

*Pseudomonas* species are metabolically robust, with capacity to produce secondary metabolites including cyclic lipopeptides (CLPs). Herein we conducted a chemical analysis of a crude CLP extract from the cocoyam rhizosphere-derived biocontrol strain *Pseudomonas* sp. COW3. We performed in silico analyses on its whole genome, and conducted in vitro antagonistic assay using the strain and purified CLPs. Via LC-MS and NMR, we elucidated the structures of four novel members of the bananamide group, named bananamides D-G. Besides variability in fatty acid length, bananamides D-G differ from previously described bananamides A-C and MD-0066 by the presence of a serine and aspartic acid at position 6 and 2, respectively. In addition, bananamide G has valine instead of isoleucine at position 8. Kendrick mass defect (KMD) allowed the assignment of molecular formulae to bananamides D and E. We unraveled a non-ribosomal peptide synthetase cluster *banA, banB* and *banC* which encodes the novel bananamide derivatives. Furthermore, COW3 displayed antagonistic activity and mycophagy against *Pythium myriotylum,* while it mainly showed mycophagy on *Pyricularia oryzae.* Purified bananamides D-G inhibited the growth of *P. myriotylum* and *P. oryzae* and caused hyphal distortion. Our study shows the complementarity of chemical analyses and genome mining in the discovery and elucidation of novel CLPs. In addition, structurally diverse bananamides differ in their antimicrobial activity.

## 1. Introduction

Fluorescent *Pseudomonas* species possess a robust metabolic machinery with the inherent ability to produce multiple and diverse secondary metabolites, including antibiotics, rhamnolipids and cyclic lipopeptides (CLPs) [1,2]. CLPs are bioactive molecules which possess multiple functions in the producing bacteria, including swarming motility, biofilm formation, virulence, and can further mediate biological control against plant pathogens via direct antagonism and elicitation of induced systemic resistance (ISR) [3,4,5,6,7]. The mode of action of CLPs is attributed to their capacity to penetrate the plasma membrane, and modify the membrane integrity of target microbes, cell and/or tissues [2,4], leading to hyphal leakage and extensive branching, among others [6,8,9,10]. *Pseudomonas* CLPs are categorized into fourteen different groups [5], based on the oligopeptide length and macrocycle size and fatty acid length. CLP members belonging to these families have been described from *Pseudomonas* strains isolated from diverse ecologies [11]. These molecules are encoded by non-ribosomal peptide synthetases (NRPSs) via distinct modules which comprise adenylation (A), condensation (C) and thiolation (T) domains [12,13,14,15].

Genome mining has led to the discovery of NRPS clusters that encode diverse CLPs [1,16]. The phylogeny of NRPS domains are quite complex and demonstrate different evolutionary patterns. The A and C domains are the most conserved and have been shown to evolve independently in the same pathway [17]. The A domain selects the cognate amino acid and generates the corresponding amino acyl adenylate. The specificity-conferring code used by the A domains has been deciphered [18], and enables structural predictions of unknown CLPs from the primary sequence. Some positions within the signature sequence are variable, allowing some flexibility in amino acid selection. On the other hand, C domain phylogeny facilitates the stereochemistry determination of the amino acids which are added to the growing peptide chain [17]. Within this phylogeny, six domain clades have been identified namely, ^L^C_L_, ^D^C_L_, Starter C, cyclization, epimerization, and dual E/C domains [19].

Chemical structures of CLPs have been largely deciphered using ultraviolet-visible (UV), infrared (IR) spectrophotometry, mass spectrometry (MS) and nuclear magnetic resonance (NMR) [20]. Novel reports of CLPs by specific strains have been accompanied by both nuclear magnetic resonance (NMR) characterization and/or genetic analyses of the NRPS gene clusters. Examples of CLPs that have been characterized using NMR, genetic analysis, or both methods include xantholysin [21], WLIP [7,22,23], viscosin [24], poaeamide [25], massetolide [26,27], lokisin [7,23], anikasin [28], orfamide [6,29], arthrofactin [30,31], putisolvin [32,33], bananamides A-C [34], cocoyamide [9], and gacamide [35], among others.

Concomitantly, chemical structure determination of certain CLPs, such as entolysin produced by *P. entomophilia* L48T, was conducted via MS, using matrix-assisted laser desorption ionization (MALDI) mass spectrometry (MS) and tandem MS (MS/MS) analysis [36]. This led to complementary results regarding CLP structure as those obtained via in silico analysis of the whole genome of L48T. Nevertheless, the use of high resolution mass spectrometry (HRMS) such as very high field Fourier transformation ion cyclotron resonance (FT-ICR) technologies or the Orbitrap and hybrid Q-TOF technologies are now commonplace.

In this context of HRMS, the Kendrick mass defect (KMD) analysis has been employed as a proof of concept for the detection, identification and discrimination of chemically related compounds on the basis of their exact masses obtained from MS [37]. The use of this method for molecular formula assignment from MS-derived exact masses is being explored for novel CLPs. The proof of concept of the KMD approach has been recently published and appears as a new tool for the elucidation of novel CLPs/CLP derivatives [38].

The first bananamide-type CLP was elucidated from *P. granadensis* F-278,770^T^ (=LMG27940^T^), which produces a bananamide derivative named MDN-0066 with the following structure, 3-OH C10:0 –Leu1–Glu2–Thr3–Leu4–Leu5–Ser6–Leu7–Ile8 [39]. In search of antitumor therapies, MDN-0066 was demonstrated to successfully induce apoptosis in renal (kidney) cancer cell lines [39]. Furthermore, another bananamide-type CLP was isolated and characterized from a banana rhizosphere isolate from Sri Lanka, *Pseudomonas* sp. BW11P2 [34]. This CLP possesses an amino acid (AA) chain length of eight with six AA in the ring. Besides the production of bananamide 1, with the chemical structure 3-OH C12:0–Leu1–Asp2–Thr3–Leu4–Leu5–Gln6–Leu7–Ile8, BW11P2 produces two other derivatives designated bananamide 2, chemically elucidated as 3-OH C10:0–Leu1–Asp2–Thr3–Leu4–Leu5–Gln6– Leu7–Ile8, and 3, chemically determined as 3-OH C12:1–Leu1–Asp2–Thr3–Leu4–Leu5–Gln6–Leu7– Ile8. We previously isolated and characterized *Pseudomonas* spp. COW3 and COW65 from the cocoyam rhizosphere in Cameroon [23]. These strains produce a common CLP designated N3 and in in vivo experiments, COW3 effectively suppressed the cocoyam root rot disease caused by *Pythium myriotylum* [9]. Recent studies showed the capacity of COW3 to induce systemic resistance against *Pyricularia oryzae* (syn. *Magnaporthe oryzae*) VT5M1 on rice, although the CLP N3 appeared not to be involved in this process [7]. However, crude N3 produced by COW3 inhibited appressoria formation and suppressed *P. oryzae* VT5M1 on rice via direct antagonism [7].

In this study, we elucidated the structure of N3 by performing chemical structural analyses, using the Kendrick mass defect-based molecular formula assignment and NMR. Also, we sequenced the genome of COW3 and conducted in silico analysis of the NRPS gene clusters. Next, we investigated the biological activity of purified novel bananamide derivatives produced by COW3 against *P. myriotylum* CMR1, a soil-borne pathogen of cocoyam (host plant of COW3 and COW65), and a foliar rice pathogen, *P. oryzae* VT5M1.

## 2. Results

### 2.1. Molecular Formula Assignment from Pseudomonas sp. COW3 Crude Extract

The crude CLP extract from *Pseudomonas* sp. COW3 was re-solubilized in and submitted to a HRMS analysis using a SYNAPT-G2-Si mass spectrometer operating in sensitivity-positive modes. The HRMS-spectra are reported in the Figure 1 and show the relative heterogeneity of extracted CLPs as protonated molecular ions [M + H]^+^ of *m*/*z* 1065.6862, 1067.6967, 1079.6995 and 1081.7089 and their sodium adducts [M + Na]^+^ respectively, *m*/*z* 1087.6946.1089.7114, 1101.7209, 1103.7072. The molecular formula of produced compounds can be deduced from the *m*/*z* (z = 1) using an approach that combines regular Kendrick mass defect calculations with knowledge stored in the NORINE database. Briefly, the m/z of a compound is represented as coordinates in the RKMD/NKM 2D-plot and the corresponding molecular formula can be deduced [37]. The exact m/z measurements were submitted to Kendrick Formula Predictor for the computer-assisted assignment of the molecular formulae of each compound, C_53_H_92_N_8_O_14_, C_53_H_94_N_8_O_14_, C_54_H_94_N_8_O_14_ and C_53_H_96_N_8_O_14_ for M^+^H^+^ 1065.6862, 1067.6967, 1079.6995 and 1081.7089, respectively (Figure 1, Table 1).

### 2.2. Chemical Analysis of Novel Bananamide Derivatives

Concentrated crude CLP from *Pseudomonas* sp. COW3 was purified by preparative chromatography. A group of four peaks (two main compounds and two minor congeners) were present (Figure 2).

These CLP peaks were subjected to further analysis using liquid state NMR spectroscopy and LC-MS spectrometry. Results revealed that all four CLP peaks consist of a 3-hydroxy fatty acid linked to a peptide chain involving eight amino acids (Figure 3, Table A1, Table A2, Table A3, Table A4). The amino acid sequences were confirmed by the analysis of the 2D ROESY and ^1^H ^13^C gHMBC spectra (Appendix A). Unfortunately, the position of the ester bond which cyclizes the molecule could not be confirmed by analysis of the ^1^H-^13^C HMBC spectra due to spectral overlap. However, the unusually high chemical shifts of Thr3 H^β^ indicated that this residue is involved in the depsi bond, indicating that the macrocycle is composed of 6 AA (Appendix A). The CLPs encoded by *Pseudomonas* sp. COW3 were designated bananamides D-G, because they are variants of bananamides A–C (or banamides 1–3) produced by *Pseudomonas* sp. BW11P2 [34], in which the Gln at position 6 is replaced by a Ser (see Table 2).

They also differ from the previously described CLP MDN-0066, with an Asp at position 2 rather than a Glu (Table 2). Bananamides D and G have an unsaturation in the fatty acid at position 5, resulting in a 3-hydroxydodeca-5-enoate fatty acid moiety. Moreover, considerations of the ^13^C chemical shifts within the lipid tail moieties suggest that the double bond is in the cis configuration (Figure 3). This is highly similar to what was found for xantholysin C [21], and corpeptin B [40]. In addition, bananamide G has valine instead of isoleucine at the eighth position of the amino acid sequence, compared to bananamide D (Figure 3, Table 2, Table A1, Table A4). The sequence determined for the main compounds (bananamides D and E) are in perfect agreement with the measured molecular formulas C_53_H_92_N_8_O_14_ (m/z 1065.6862), and C_53_H_94_N_8_O_14_ (m/z 1067.6967) using Kendrick Formula Predictor (see Table 1). The two minor congeners could not be picked up by the Kendrick Formula Predictor.

### 2.3. In Silico Analysis of the NRPS Synthetases Encoding the New Bananamide Derivatives

The whole genome of *Pseudomonas* spp. COW3 was analyzed and compared with the genome of *Pseudomonas* sp. COW65, a related strain from the cocoyam rhizosphere that also produces the N3-type CLP [23]. Genome mining results revealed that the new bananamide derivatives are encoded by NRPS synthetases consisting of an adenylation (A), condensation (C) and thiolation (T) domain including two TE domains situated at the end of the biosynthetic gene. The NRPS gene cluster comprises three genes, namely, *banA* (two modules, 6.4 kb), *banB* (four modules, 13 kb) and *banC* (8.3 kb) (Figure 4). Compared to the gene cluster of the bananamide producer BW11P2, a *luxR*-type transcriptional regulator gene (*banR1*) and *nodT*-like outer membrane lipoprotein gene (*banT*) are situated upstream of *banA,* while *banC* is flanked downstream by two transporter genes which encode macrolide efflux proteins MacA (*banD*), MacB (*banE*), and an additional *luxR*-type transcriptional regulator gene (*banR2*) (Figure 4).

In NRPSs, adenylation (A) domains recruit and select amino acids for peptide biosynthesis [14]. Subsequent in silico analysis of the A domain substrate using antiSMASH v5.0 allowed prediction of the amino acid composition of the peptide moiety. A phylogenetic analysis was carried out with the A domains retrieved from COW3, COW65, the bananamide A-C producer BW11P2, the MDN-0066 producer LMG 27940 and putative bananamide-producers, including *Pseudomonas* sp. MS586, MS82, DR 5-09, BS3668, Z003-0.4C (8344-21) and R45 (see Table A5). Our results showed that similar amino acids as those of COW3 may be recruited by COW65, while BS3668 and Z003-0.4C (8344-21) may recruit similar amino acids as the MDN-0066 producer LMG 27940. More so, strains MS586, MS82, DR 5-09 and R45 may recruit the same amino acids as BW11P2 which suggests that these four strains are likely bananamide A-C producers (Appendix A). The predicted amino acid sequence of *Pseudomonas* sp. COW3 CLP is Leu1-Asp2-Thr3-Leu4-Leu5-Ser6-Leu7-Ile8 (Figure 4 and Appendix A), which was confirmed by the chemical analysis (Table 2, Figure 3). Our results confirm that the amino acids assembled by the NRPSs of COW3 and COW65, are closely related to those of the previously reported bananamide A-C producer, BW11P2 and the MDN-0066 producer LMG 27940, but differ from bananamide A-C at the sixth position with the selection of Ser instead of Gln. They also differ from MDN-0066 at the second position with the selection of Asp instead of Glu (Table 2, Appendix A, and Figure 4). Analysis of the C domains of each module in the bananamide D gene cluster identified a lipo-initiation (C_starter_) domain in module 1 of *banA*. Further C domain analysis predicted module G to be ^L^C_L_ domains while the other C domains were classified as C/E domains (Appendix A). The predicted isomery (d- and l- configuration) of the amino acid sequences of bananamide D and other (putative) bananamide and MDN-0066-producing strains is d-Leu1, d-Asp/Glu2, d-Thr3, d-Leu4, d-Leu5, d-Ser/Glu6, l-Leu7, Ile/Val8 (Appendix A). Although chemical analysis is required to validate the predicted isomery, it could serve as a clue to elucidating the chemical configuration. Further in silico analysis showed that the first Te domain clusters with the Type 1 Te domains of *Pseudomonas* CLPs which function in the cyclization of the C-terminal amino acid via a threonine or serine residue, while the second Te domain forms a clade with type II Te domains with functions in the support of the assembly line (Appendix A). Protein sequence comparison with Blastp showed a very high identity (95–98%) of COW3 NRPS proteins with those of COW65, the putative bananamide D-G-producing strain which is also associated with the same host. A somewhat lower protein identity was found with the synthetases of the MDN-0066 producer LMG 27940 (74–79%) and the putative MDN-0066 producers BS3668 (73–80%), and Z003-0.4C(8344-21) (73–79%). Furthermore, COW3 NRPS proteins were not closely identical with those of the bananamide A-C producer BW11P2 (80-84%), and other putative bananamide A-C producers such as *Pseudomonas* sp. DR5-09 (80–85%), *Pseudomonas* sp. MS586 (80–84%), *Pseudomonas* sp. MS82 (80-85%) and *Pseudomonas* sp. R45 (80–85%) (Appendix A). Consequently, a cladogram inferred from the concatenation of regulatory, export and NRPS gene protein sequences clearly delineated the novel bananamide D-G, bananamide A-C and MDN-0066 into these three distinct members of the bananamide group including the putative CLP-producing strains (Figure 5). The NPRS biosynthetic gene cluster and flanking regions for *Pseudomonas* spp. COW3 and COW65 have been deposited on Genbank with accession number MN480426.

### 2.4. Phylogenetic Analysis of Pseudomonas spp. Based on rpoD, rpoB, gyrB and 16S rDNA Gene Sequences

Results from the phylogenetic MLSA analyses using house-keeping genes of CLP-producing *Pseudomonas* strains show that all the bananamide-producing strains belong to the *P. koreensis* group and cluster with the type strain *P. koreensis* LMG 21318^T^. *Pseudomonas* sp. COW3, which produces bananamides D-G clustered with the putative bananamide D-G-producer COW65, separately from both known and putative bananamide A-C and MDN-0066 producers, respectively (Figure 6). The phylogenetic MLSA analyses of the bananamide producers is in agreement with the phylogenetic analysis of the A domains from the CLPs biosynthetic genes (Appendix A) and the cladogram presented in Figure 5.

### 2.5. In Vitro Antagonistic Activity of Pseudomonas sp. COW3 and Purified Bananamide D-G against P. Myriotylum

Antagonistic activity of COW3 against *P. myriotylum* was evaluated using co-culturing tests. COW3 strongly inhibited the growth of *P. myriotylum* CMR1 with clear zones of inhibition after incubation for two and four days, respectively in two independent experiments (Figure 7A,B). Subsequent microscopic analysis using purified bananamide D, E, F and G from COW3 showed antagonistic activity against *P. myriotylum* CMR1 between 10 to 50 µM, with the exception of bananamide E which was also active at 1 µM (Figure 8A). Microscopic analysis of the interaction of the pure CLPs with *P. myriotylum* CMR1 (Figure 8B) revealed the capacity of all bananamide variants to cause hyphal branching in *P. myriotyum* CMR1. In addition to hyphal leakage observed at 1µM for bananamide D and at 25 µM with bananamide E, F and G, bananamide D caused vacuolization at 1 to 25 µM while bananamide E led to vacuolization and hyphal curling at 10 µM (Figure 8).

### 2.6. In Vitro Antagonistic Activity of Pseudomonas sp. COW3 and Purified Bananamide D-G against P. Oryzae

Antagonistic activity of *Pseudomonas* sp. COW3 on *P. oryzae* evaluated using the co-culturing test, showed no inhibitory effect on *P. oryzae* VT5M1 and Guy11 after four days of co-culturing but a strong predatory potential on the fungi was displayed after incubation for seven days (Figure 9). Microscopic analysis with purified bananamide D-G from COW3 revealed growth inhibition, extensive branching, hyphal leakage, and vacuolization of the mycelia of *P. oryzae* GUY11 and VT5M1, at concentrations ranging from 1 to 50 μM. Furthermore, structure-function activity was also displayed in the antagonism of purified bananamides D-G against *P. oryzae* VT5M1 and GUY11. Bananamide D caused the strongest growth inhibition on both strains in a dose dependent manner, while 10 µM of bananamide E gave a higher effect on GUY11 than 25 and 50 µM, which had activities comparable to the other variants. In addition to hyphal branching caused by all CLP variants, 25 µM of bananamide D and 10 µM of bananamide G, led to hyphal leakage of GUY11, while only bananamide D caused hyphal leakage of VT5M1. Hyphal vacuolization was caused by 1 and 50 µM of bananamide D and G on GUY11 and VT5M1, respectively (Figure 10).

## 3. Discussion

In this study we conducted the chemical characterization, genetic analysis and biological activity of novel bananamide derivatives, bananamides D-G, produced by *Pseudomonas* sp. COW3. A combination of genome mining and subsequent chemical analysis enabled the identification of the major CLPs produced by COW3.

Using NMR, we were able to identify two major CLPs, designated bananamide D and E, in combination with two minor derivatives called bananamides F and G. Chemical analysis revealed that these compounds are similar to the recently described bananamides [34], MDN-0066 [39], and the pseudofactins [41]. However, bananamide D and variants produced by *Pseudomonas* sp. COW3 differ from the previously described CLPs by amino acid substitution at certain positions, fatty acids tail and/or level of saturation. This is a common phenomenon with CLP-producing bacteria such as *Bacillus* spp., *Streptomyces* spp. and *Pseudomonas* spp. owing to their capacity to harness and incorporate amino acids in the environment to enable CLPs synthesis, thereby conferring a competitive advantage to them in relation to other microorganisms in the environment. For bananamide D and derivatives, the substitution of an amino acid at the eighth position, disparity in the length and presence of unsaturation at the fifth position in the fatty acid led to the confirmation of the novelty of the four structurally diverse bananamide CLPs from *Pseudomonas* sp. COW3 strain. The Kendrick mass defect method for molecular formula assignment has been used for the detection and discrimination of chemically related compounds based on their exact masses [37]. In this study, we present the first use of KMD in the discovery of novel bananamide variants. KMD identified the two major bananamides produced by *Pseudomonas* sp. COW3 (D and E). KMD approach for molecular formula assignment to CLPs is therefore an effective and reliable method for the determination of the molecular formulae of CLPs using the experimentally determined high resolution mass (HRM). The HRM is often compared to masses of CLPs on NORINE database, thereby enabling the discovery and identification of novel CLPs.

Consequently, bioinformatics analyses revealed the eight amino acid composition of CLP produced by COW3 as follows: FA-D-Leu-D-Asp-D-Thr-D-Leu-D-Leu-D-Ser-L-Leu-L-Ile, which is consistent with the “colinearity rule” of nonribosomal peptides and chemical analysis. We discovered that bananamide D-G are novel bananamide derivatives, which differ from MDN-0066 (P. granadensis F-278,770T) and bananamides A-C (BW11P2) in the amino acids composition. Furthermore, our in silico analysis showed that similar to the NRPS backbone of the bananamide producer, BW11P2, our strain (COW3) possesses three NRPS genes *banA, banB* and *banC*. These results demonstrate that genome mining remains a powerful tool for the discovery of novel metabolites including CLPs [16].

In order to obtain insights into the biological activities of COW3, we conducted whole cell co-cultivation assay against the soil-borne cocoyam pathogen, *P. myriotylum* CMR1, and the rice blast pathogens, *P. oryzae* VT5M1 and Guy11. In this study, whole cells of COW3 showed in vitro antagonistic activity against *P. myriotylum* CMR1 with clear zones of inhibition after two days, coupled with mycophagy after four days of co-incubation with the pathogen. A recent study reports that *Trichoderma* strains with in vitro antagonism against *P. myriotylum* all produced cellulases, proteases, and xylanases [42]. It remains to be investigated whether this is also the case for our bacterial strain *Pseudomonas* sp. COW3. Our results are interesting considering the origin and context in which COW3 was isolated. COW3 is a cocoyam rhizosphere isolate obtained from the roots of cocoyam plants grown in a *Pythium* root rot disease-suppressive Boteva soil [9]. Our findings suggest that COW3 is a likely player, alongside other CLP producers, in carving a ‘safe haven’ for its cocoyam host plant.

On the other hand, in co-cultivation experiments with *P. oryzae* VT5M1 and Guy11, COW3 mainly mediates hyphal digestion and mycophagy after seven days. Strains with biocontrol capacities have been reported to demonstrate mycophagous behaviour against fungal plant pathogens via synthesis of proteases, chitinases, N-acetyl-beta-d-glucosaminidase and glucanases [43]. The functional role of CLPs in direct antagonistic and mycophagous activities of COW3 needs to be studied further using site-directed mutagenesis.

In previous studies, crude bananamide D-G (previously named CLP N3) inhibited appressoria formation by *P. oryzae* and reduced rice blast disease severity [7]. Besides the sensitivity of renal carcinoma cell lines to MDN-0066 [39], there has been no specific study aimed at investigating the biological activity of bananamides. In this study, we investigated the effect of structural diversity on the activity of purified bananamide D-G against our target pathogens, *P. myriotylum* CMR1 and *P. oryzae*. The goal was to decipher the effect of substitution of a valine for isoleucine at position 8 in bananamide G and to further investigate how differences in the fatty acids residue and level of saturation of bananamide D-G could affect their biological activity against these pathogens.

Our result showed that pure bananamide D, E, F and G produced by *Pseudomonas* sp. COW3 displayed antagonistic activity against *P. myriotylum* CMR1 which could be attributed to the detrimental effect of CLPs on the oomycete. Also, microscopic analysis revealed the capacity of bananamide D-G to cause hyphal branching in *P. myriotyum* CMR1. It is known that CLPs act by compromising the membrane integrity of phytopathogens resulting in anti-oomycete activities, among others [4,19,44]. This is in agreement with the observation of hyphal branching in *P. myriotylum* NGR03 due to purified xantholysin, putisolvin, entolysin and white line inducing principle (WLIP) [9]. Similar results of hyphal branching were observed with *R. solani* AG 4-HGI and AG2-1, in interaction with orfamides [3,6], and viscosinamide-mediated branching in *R. solani* [4,45]. Furthermore, structure-function activity was also displayed in the antagonism of purified bananamide D-G against *P. oryzae* VT5M1 and GUY11. Treatments with bananamide D resulted in higher inhibition zones in both genetically different isolates, in a dose dependent manner. In comparison with other minor variants, bananamide E appeared to have a higher inhibitory efficacy on *P. oryzae* at 10 µM. In addition to hyphal branching by the CLP variants, variable concentrations of bananamide variants produced by COW3 displayed hyphal vacuolization. It has been shown that increase in the number of carbon atoms in β-amino fatty acid chain of iturin-type lipopeptides produced by *Bacillus amyloliquefaciens* SD-32, enhanced biological activity against fungal pathogens [46]. More so, a decrease in defense-inducing activity was observed after the metabolic engineering of *Bacillus* lipopeptides involving various substitutions of Val/Leu, Leu/Val, Leu/Ile and Val/Ile in the peptide moiety. Reduction in the length of fatty acid chains of these lipopeptides from 14 carbons, to 12 or 13 carbons led to loss of activity [47]. However, the substitution of valine with isoleucine at position 4, and difference in the length of fatty acid chain had no effect on the activity of orfamides on *Rhizoctonia solani* AG 4-HGI [6]. Thus, our findings corroborate earlier reports by [46,47], that structural changes/differences in a lipopeptide could influence its biological activity. Furthermore, our result showed that bananamide E, which has a 3OH-C12:0 lipid tail, caused hyphal curling in *P. myriotylum* CMR1 unlike bananamide D, F and G. Bananamide G with valine at position eight triggered vacuolization in *P. oryzae* VT5M1 mycelia. Bananamide D with Ile8 displayed higher inhibition and mycelial distortion in comparison to bananamide G with the same fatty acid tail (3OH - C12:1), but Val8. Based on this, we therefore posit that specific amino acids at certain position(s) and length of fatty acids could influence the structural-function activity of *Pseudomonas*-derived CLPs.

Phylogenetic MLSA analyses using *rpoD, gyrB, 16S rDNA* and *rpoB* partial sequences further positioned all bananamide-producers and (novel) derivatives in the *P. fluorescens* complex - a conglomerate group comprising nine separate groups whose members are characterized by enormous secondary metabolite diversity including CLPs [48,49]. Specifically, the bananamide producers are situated in the *P. koreensis* group - a group which also situates strains producing CLPs such as lokisin, amphisin, cocoyamide and other amphisin group members. Thus, the discovery of bananamide D-G and putative bananamide producers further expands the metabolic reputation of the *P. koreensis* group and at large, that of the *P. fluorescens* complex. The result of the phylogenetic analysis of the housekeeping genes is in perfect agreement with the phylogenetic analysis based on NRPS protein sequences and A domains and chemical analyses. Continuous genome mining of secondary metabolites will further unravel what novel chemistry nature has in store.

The diverse origins of bananamide-like producers are intriguing. Bananamide producers appear to be principally associated with the rhizosphere of plants in the tropical regions of the world which are characterized by high humidity (rainy season) and dry periods (dry season). *Pseudomonas* spp. COW3 and COW65 were isolated from the cocoyam rhizosphere in tropical rain forests of Cameroon [9]. Moreover, the bananamide A-C producer, *P. fluorescens* BW11P2, was isolated from the banana rhizoplane in the tropical wetlands of Galgadera, Sri Lanka [34,50], while the putative bananamide-producing strain *Pseudomonas* sp. R45 was isolated from the sugar cane rhizosphere soil sample in Piracicaba-SP, Brazil [51,52]. Other putative-bananamide producing strains *Pseudomonas* spp. DR 5-09, MS586 and MS82 were isolated from plant roots at Daecheong, South Korea, a cotton field in the USA and from soybean roots in Mississippi, USA, respectively. The MDN-0066 producer *P. granadensis* F-278,770T originates from a soil sample obtained from the Tejeda, Almijara and Alhama Natural Park, Granada, Spain [39,53]. In contrast, the two putative MDN-0066 producers, Z003-0.4C(8344-21) and BS3668 appear to be associated with entirely different ecologies. The isolation metadata on the Genomes Online Database reports that Z003-0.4C(8344-21) is a human-associated strain which was collected from the human airways (respiratory system); BS3668 was isolated from a waste treatment plant in Idaho, USA. The occurrence of these CLPs in different hosts and diverse ecologies suggest that these compounds can demonstrate various ecological roles.

## 4. Materials and Methods

### 4.1. Strains, Media and Growth Conditions

*Pseudomonas* and *P. oryzae* strains used in this study are presented in Table 3. *Pseudomonas* strains were cultured on King’s B (KB) agar [54] at 28 °C. *P. myriotylum* CMR1 [55] was grown on glucose casamino acid and yeast extract agar (GCY) [56]. *P. oryzae* isolates VT5M1 [57] and Guy11 [58] were grown on complete medium (CM) [59], at 28 °C for five days.

### 4.2. Extraction of Crude CLPs from Pseudomonas sp. COW3

Crude CLPs were extracted from *Pseudomonas* sp. COW3 using an established protocol [7,9].

### 4.3. Crude CLP Analysis of COW3 via the Kendrick Mass Defect Method for Molecular Formula Assignment

Crude CLP extract from *Pseudomonas* sp. COW3 was re-solubilized in methanol and directly injected at 15 µL/min in a SYNAPT-G2-Si mass spectrometer (Waters, Manchester, UK), operating in positive and sensitivity modes. The sample was electrosprayed at an applied voltage of 3 kV, using a desolvation gas (N_2_) flow of 500 L/h, a nebuliser gas flow of 6.5 bar and desolvation temperatures of 150 and 300 °C, respectively. The full MS survey scans were acquired at a 20,000 (FWHM) resolving power, over the mass range of 200–2000 m/z. Peaks were analyzed using Mass Lynx software (ver.4.1; Waters). Mass spectra were externally calibrated in positive mode using a solution of 5 mM sodium formate (Sigma-Aldrich, Saint-Quentin-Fallavier, France) in 50/50 (*v*/*v*) acetonitrile (ACN)/H_2_O.

### 4.4. Molecular Formula Assignment with Kendrick Mass Defect

The Kendrick mass (KM) related to the CH_2_ pattern is calculated from the molecular formula of NORINE-referenced compounds and from the experimentally measured masses. A dealiasing, which corresponds to a mathematical shift that leads to the mass to fit the spectral width without aliasing, was performed by shifting the KM value by −0.28, equation according to [38]:
(1)KM=(IUPAC protonated monoisotopic mass from NORINE or experimentally measured mz×(CH2 nominal massCH2 exact mass))−0.28

Subsequently, the dealiased KM value is rounded to the nearest whole number and defines the nominal Kendrick mass (NKM), equation:(2)Nominal Kendrick mass (NKM)= dKM rounded to the nearest whole number

The NKM is then subtracted from Kendrick mass to obtain the regular Kendrick mass defect (RKMD), equation:(3)Regular Kendrick mass defect (RKMD)= KNM−dKM

RKMD and the NKM values were calculated for each known and curated molecular formulae extracted from the NORINE database. Subsequently, a RKMD/NKM 2D-plot (without aliasing) was generated using these values. A vectorial mesh enables the assignment of the molecular formulae and a web application (named Kendrick Formula Predictor) implementing the method can be run on-line with any list of mass-to-charge ratios at the following URL: http://bioinfo.cristal.univ-lille.fr/kendrick-webapp/.

### 4.5. Analytical and Preparative Liquid Chromatography (LC) from Crude Extract

Using crude CLP extract from *Pseudomonas* sp. COW3, analytical LC-MS data were collected on a 1100 Series HPLC (Agilent Technologies, Santa Clara, CA, United States) equipped with an ESI ionisation source, coupled to a type SL MS detector. The HPLC was equipped with an analytical Kinetex C18 reversed-phase column (150 × 4.6 mm, 5 μm particle size; Phenomenex, Torrance, CA, USA). An elution gradient of H_2_O/CH_3_CN (25:75 to 0:100, *v*/*v*) over 20 min was applied at a flow rate of 1 mL.min^-1^. The crude CLP extract was dissolved in methanol and the purification conditions of each CLP were optimized by using an Agilent Technologies 1100 Series HPLC device equipped with a Luna C-18 analytical reversed phase HPLC column (250 × 4.6 mm, 5 μm). The signal was detected using a diode array detector at a wavelength of 214 nm. Different gradient elutions of H_2_O/CH_3_CN were tested to find the optimal separation conditions while the column temperature was kept at 35°C. Large scale purification of CLPs was subsequently performed by injection of the methanol solution into a Prostar HPLC device (Agilent Technologies) equipped with a Luna C-18(2) preparative RP-HPLC column (250 × 21.2 mm, 5 μm particle size) for separation of the individual CLP analogues. The optimal elution gradient (25:75 to 0:100, *v*/*v*) of H_2_O/CH_3_CN was applied over 20 min at a flow rate of 17.5 mL.min^-1^, while the column temperature was kept at 35 °C.

### 4.6. Chemical Characterization of Novel Bananamide Derivatives via Nuclear Magnetic Resonance (NMR)

All NMR measurements were performed on an Avance III spectrometer (Bruker, (Billerica, MA, USA) operating at a respective ^1^H and ^13^C frequency of 500.13 MHz and 125.76 MHz equipped with a BBI-Z probe. The sample temperature was set to 298.0 K. Standard pulse sequences as present in the Bruker library were consistently used unless otherwise stated. Spectra that were used for NMR assignment included ^1^H-^1^H COSY (cosygpqf), ^1^H-^1^H TOCSY (mlevph), ^1^H-^1^H NOESY (noesygpph),^1^H-^1^H ROESY (offroesygpprphpp), ^1^H-^13^C HSQC (hsqcedetgpsisp2.4) and ^1^H-^13^C HMBC (hmbcgplpndqf). High precision 5 mm NMR tubes (Norell, Landisville, NJ, USA) were used. Dimethylformamide-d7 (DMF) (99.50%) was purchased from Eurisotop (Saint-Aubin, France). ^1^H and ^13^C chemical shift scales were calibrated by using the residual solvent signal employing Tetramethylsilane (TMS) as secondary reference.

2D spectra measured for structure elucidation include a 2D ^1^H-^1^H DQF-COSY, 2D ^1^H - ^1^H TOCSY with a 90 ms MLEV-17 spinlock, 2D ^1^H-^1^H off-resonance ROESYs with 200 ms mixing time and gradient-selected ^1^H-^13^C gHSQC and gHMBC. Typically, 2048 data points were sampled in the direct dimension for 512 data points in the indirect dimension, with the spectral width set to 11 ppm and 110 ppm along the ^1^H and ^13^C dimension, respectively. The ^1^H-^13^C HMBC was measured with a 210 ppm ^13^C spectral width. For 2D processing, the spectra were zero filled to a 2048 × 2048 real data matrix. Prior to Fourier transformation, all spectra were multiplied with a squared cosine bell function in both dimensions or sine bell in the direct dimension for the gHMBC.

### 4.7. Genome Sequencing and Assembly

For genome sequencing, *Pseudomonas* sp. COW3 (N3 producer) and COW65 (a putative N3 producer, [9]), were grown in Luria Bertani (LB) broth for 24 h at 28 °C with continuous shaking at 150 rpm. Genomic DNA was extracted using the Wizard Genomic DNA Purification Kit (Promega Corporation, Madison, WI, USA) according to manufacturer’s instructions.

Single-end or paired-end sequence reads were generated using the Illumina HiSeq2500 or MiSeq system at the BASECLEAR B. V. (Leiden, The Netherlands). FASTQ read sequence files were generated using bcl2fastq2 version 2.18. Initial quality assessment was based on data passing the Illumina Chastity filtering. Subsequently, reads containing PhiX control signal were removed using an in-house filtering protocol. In addition, reads containing (partial) adapters were clipped (up to a minimum read length of 50 bp). The second quality assessment was based on the remaining reads using the FASTQC quality control tool version 0.11.5 (Babraham Institute, Cambridge, UK)

The quality of Illumina reads was improved using the error correction tool BayesHammer [60]. Error-corrected reads were assembled into contigs using SPAdes version 3.10 [61]. The order of contigs, and the distances between them, was estimated using the insert size information derived from an alignment of the paired-end reads to the draft assembly. Consequently, contigs were linked together and placed into scaffolds using SSPACE version 2.3 [62]. Using Illumina reads, gapped regions within scaffolds were (partially) closed using GapFiller version 1.10 [63]. Finally, assembly errors and the nucleotide disagreements between the Illumina reads and scaffold sequences were corrected using Pilon version 1.21 [64].

### 4.8. Genome Annotation, Genome Mining and Bioinformatics Analyses

Genome sequences were automatically annotated using the RAST annotation pipeline [65,66]. Furthermore, genomes of previously sequenced strains (Appendix A) were re-annotated using the RAST annotation pipeline and also submitted to antiSMASH v5.0 [67]. Genome mining was conducted on the annotated genomes, and comparison of NRPS proteins with other protein sequences in GenBank database was done by BLAST search (https://blast.ncbi.nlm.nih.gov/Blast.cgi). The adenylation (A) and condensation (C) domains sequences of the non-ribosomal peptide synthase (NRPS) genes were extracted. Sequence alignment was carried out using MUSCLE [68] in the software package Geneious Prime, 2019, and the cladograms were inferred by Neighbour Joining. The thioesterase (TE) domain sequences of the non-ribosomal peptide synthase (NRPS) gene were extracted. Sequence alignment was carried out using MUSCLE [68], and the phylogenetic tree was constructed by maximum likelihood with 1000 bootstrap replication, in the software package MEGA6 [69]. The antiSMASH v5.0 enabled the prediction of the amino acid composition of the peptide moiety. Moreover, proteins were translated from the *nrps* and flanking genes of bananamide D-G-and other CLP-producing strains (Appendix A). The protein sequences were concatenated, aligned and used to construct a phylogenetic tree in Geneious Prime, 2019 (Biomatters, Auckland, New Zealand). 

### 4.9. Gap Filling of NRPS Scaffold Sequences in Pseudomonas sp. COW3

Genomic DNA was extracted using the Wizard Genomic DNA Purification Kit from Promega. The region of interest was amplified by PCR using primers: COW3G_F (GTTCGGTTTCGATGCGATGG) and COW3G_R (GATTTCCATTTCGCCGACCG) designed with Geneious 11.1.5 software (https://www.geneious.com). PCR reaction was carried in a 50 µL mixture containing 10 µL 5× Go Taq Reaction buffer, 0.2 µL Go Taq DNA Polymerase (Promega), 1 µL of 10 mM dNTP mixture, 2 µL of 10 µM COW3G_F primer, 2 µL of 10 µM COW3G_R primer, 2 µL genomic DNA, and 32.8 µL double-distilled water. Amplification was performed using a FlexCycler Block Analytikjena 2010 with the following program: 94 °C for 2 min, 30 cycles of 94 °C for 30 s, 65 °C for 1 min, 72 °C for 1 min, and extension at 72 °C for 10 min, and finally kept at 4 °C. Amplicons were visualized by 1.5% gel electrophoresis after staining with ethidium bromide. Purified PCR product was sequenced at LGC Genomics GmbH (Berlin, Germany).

### 4.10. Phylogenetic Analyses of Pseudomonas Strains

For multi-locus sequence analyses (MLSA), *rpoD, rpoB,* 16S rDNA and *gyrB* sequences of our bananamide D-G-producing *Pseudomonas* strains (COW3 and COW65) were extracted from their corresponding draft genome sequences. In addition, draft genomes of putative bananamide-producing strains, and sequences of selected *Pseudomonas* type strains were retrieved from GenBank (Table A5). The sequences were aligned using MUSCLE [68] in MEGA6 [69]. A phylogeny tree was constructed by maximum likelihood with 1000 bootstrap replication, and *P. aeruginosa* was used as outgroup. *rpoD, rpoB, 16S rDNA* and *gyrB* trees were generated separately after which a concatenated tree was obtained by combining the aligned sequences of the four genes.

### 4.11. In Vitro Direct Antagonism of Pseudomonas sp. COW3 against P. myriotylum CMR1

In vitro antagonism of the bananamide-producing strain COW3, against the soil-borne oomycete *P. myriotylum* CMR1, which causes the cocoyam root rot disease, was investigated by Petri dish assay, as described by [70]. Three (3) µL of overnight King B [54] broth culture of the test bacteria were spotted on either side of GCY agar plates spanning 2 cm from the center. The plates were incubated for 24 h at 25 °C, after which a mycelium plug of CMR1 (5 mm in diameter) was placed at the center of the plate. The plates were incubated at 28 °C, and pictures were taken after two and four days.

### 4.12. In Vitro Antagonism of Pseudomonas sp. COW3 against P. oryzae

The Petri dish assay was used to investigate the in vitro antagonism of COW3, against two genetically different isolates of the rice blast pathogen, *P. oryzae* VT5M1 and *P. oryzae* Guy11, as described by [70]. Three (3) µL of overnight LB broth culture of the test bacteria were spotted on either side of complete medium (CM) plates spanning 2 cm from the center. The plates were incubated for 24 h at 25 °C, after which a mycelium plug of VT5M1 or Guy11 (5 mm in diameter) was placed at the center of the plate. Plates were re-incubated at 25 °C for 4 to 10 days.

### 4.13. In Vitro Microscopic Inhibition of *P. myriotylum*, and *P. oryzae* using Purified Bananamides D-G

The antagonistic activity of purified bananamides D-G was studied, in interaction with *P. myriotylum* CMR1, and *P. oryzae* VT5M1 and Guy11. The experiment was conducted in in vitro microscopic conditions as described by [9], with minor modifications. Sterile microscopic glass slides were covered with a thin film of water agar (Bacto agar; Difco) and placed in a plastic Petri dish containing sterile filter paper moistened with 2 mL of sterile distilled water. A single agar plug (diameter = 5 mm) of *P. myriotylum* CMR1 and *P. oryzae* isolates (VT5M1 and Guy11) was obtained from a five day old culture grown on GCY agar and CM agar, respectively. Plugs were inoculated at the center of each glass slide. Stock solutions (5 mM) of bananamides D, E, F and G were initially made by dilution with DMSO while subsequent dilutions were made using sterile MilliQ water. Fifteen (15) µL of each bananamide CLP solution at concentration 1 µM, 10 µM, 25 µM and 50 µM, was added separately at either side of the glass slide (2 cm from the fungal plug). *P. myriotylum* CMR1 assay was incubated for 3 days, while *P. oryzae* plates were incubated for six days at 28 °C. Microscopic slides were assessed for hyphal distortion phenotypes under an Olympus BX51 microscope. The efficacy of treatments of each bananamide variant was evaluated by measurement of mycelial growth diameter compared with those obtained with the *P. myriotylum* and *P. oryzae* control. Percentage inhibition of the fungal pathogens by bananamide variants and for each concentration was calculated relative to the mycelial growth in the control. Percentage inhibition was calculated according to the following formula:$$\frac{\left(Growth\;diameter\;untreated\;control-Growth\;diameter\;treated\;control\right)x100}{growth\;diameter\;untreated\;control}$$

Data obtained were analyzed using SPSS 25 statistical software. To compare across treatments, univariate ANOVA followed by Tukey’s posthoc tests, were used. Results were considered to be statistically different when *p* < 0.05. Figures were generated to represent the percentage inhibition of the fungi. Representative pictures of mycelial damage due to CLP-pathogen interaction are shown.

## 5. Conclusions

This study shows that the KMD approach [38] can be employed for the discovery of new CLP. However, for a full chemical characterization, NMR remains a necessary option as shown for the characterization of bananamide D, E, F and G, in combination with genetic analysis. The result obtained from the phylogenetic analyses of the housekeeping genes and protein sequences of the NRPS and flanking regions, complements the genetic and chemical analyses further confirming the novelty of bananamides D-G. We further showed that the antagonism of COW3 towards *P. myriotylum* CMR1 involves both direct inhibition and mycophagy while the antagonistic activity of COW3 against *P. oryzae* mainly involves mycophagy. Our study showed the influence of structural diversity in amino acid and fatty acid composition on the biological activity of bananamides D-G. It remains to be investigated, however, whether bananamides play a role in mycophagy.

## Figures and Tables

**Figure 1 molecules-24-04170-f001:**
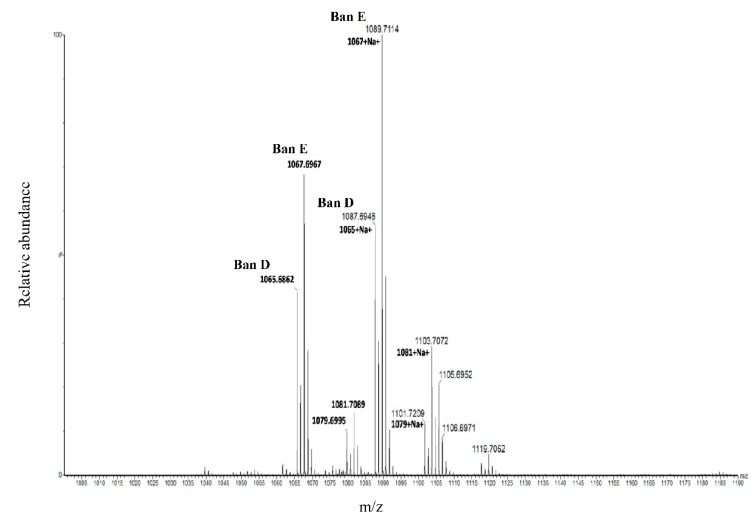
Kendrick mass defect molecular formular assignment of crude CLP extract from *Pseudomonas* sp. COW3. ESI QTOF mass spectrum in the mass range 1000-1190 m/z obtained from the direct introduction of crude CLP extract from COW3.

**Figure 2 molecules-24-04170-f002:**
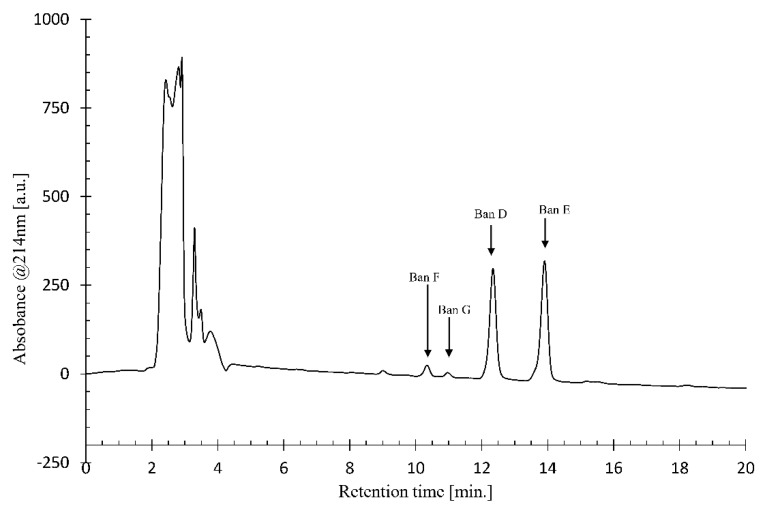
Chemical analysis of crude CLP samples from *Pseudomonas* sp. COW3. Representative preparative HPLC chromatogram of the crude CLP extract. CLPs analysed are indicated with arrows. Gradient: 25:75 to 0:100 H_2_O: acetonitrile in 15 min.

**Figure 3 molecules-24-04170-f003:**
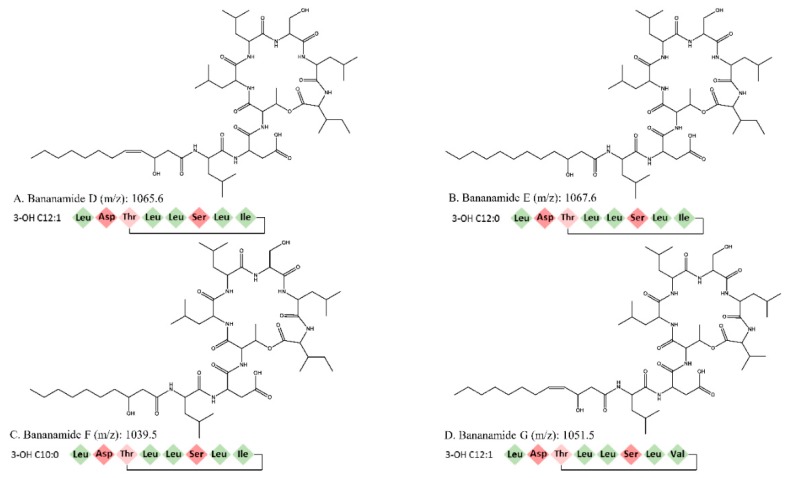
Chemical structure of bananamide D, E, F and G, with m/z values and corresponding CLP sequences showing amino acids involved in cyclization. Hydrophobic amino acids are indicated in green, hydrophilic amino acids in pink.

**Figure 4 molecules-24-04170-f004:**
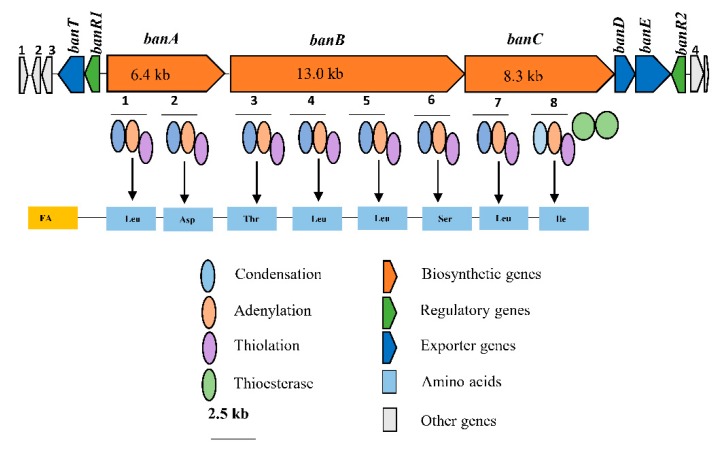
In silico analyses of NRPS gene clusters encoding bananamides D–G. Gene clusters comprising three structural NRPS genes, designated *banA*, *banB* and *banC* responsible for bananamide D-G biosynthesis in COW3. Detailed comparison of bananamide D-synthetases and flanking gene protein sequences are shown in Appendix A. Each module contains an adenylation (A), condensation (C) and thiolation (T) domains, and two TE domains for peptide release and cyclization. Scale bar represents 2.5 kb.

**Figure 5 molecules-24-04170-f005:**
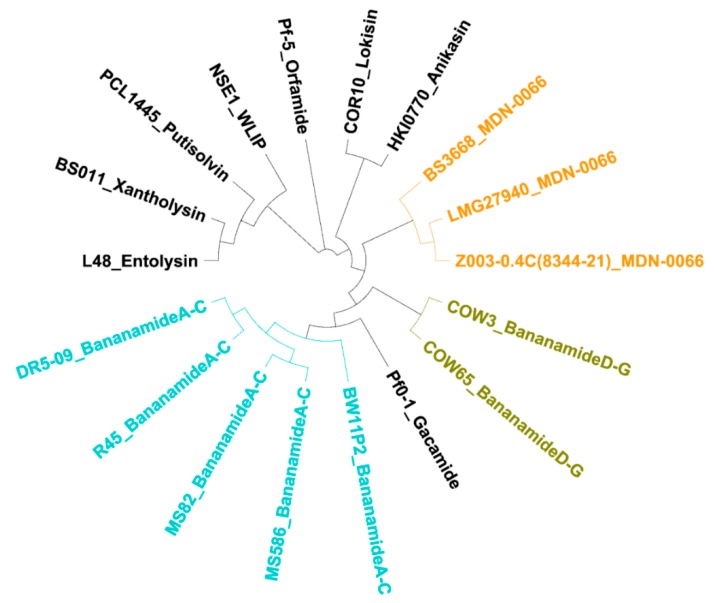
Phylogenetic analysis of concatenated regulatory, export and biosynthetic protein sequences extracted from bananamide D-G, and from other characterized and putative CLP-producing strains. Cladogram of neighbor joining tree inferred from the alignment of concatenated RND efflux system, outer membrane lipoprotein CmeC (NodT), transcriptional regulator, LuxR family (LuxR up and down downstream), non-ribosomal peptide synthetases (NRPSABC), macrolide-specific efflux protein (MacA) and macrolide export ATP-binding/permease protein (MacB) protein sequences, extracted from already characterized and putative *Pseudomonas* cyclic lipopeptides NRPSs including novel bananamide D-G (*Pseudomonas* sp. COW3 and *Pseudomonas* sp. COW65); bananamide A-C (*Pseudomonas* sp. BW11P2); putative bananamide A-C (*Pseudomonas* sp. MS586, *P. fluorescens* MS82, *Pseudomonas* sp. R45 and *Pseudomonas* sp. DR5-09); MDN-0066 (*P*. *granadensis* LMG 27940); putative MDN-0066 (*Pseudomonas* sp. Z003-0.4C(8344-21) and *P. moraviensis* BS3668); entolysin (*P. entomophila* L48); xantholysin (*P. mosselii* BS011); putisolvin (*P. putida* PCL1445); gacamide (*P. fluorescens* Pf01); WLIP (*Pseudomonas* sp. NSE1); lokisin (*Pseudomonas* sp. COR10); anikasin (*P. fluorescens* HKI0770) and orfamide (*P. protegens* Pf-5).

**Figure 6 molecules-24-04170-f006:**
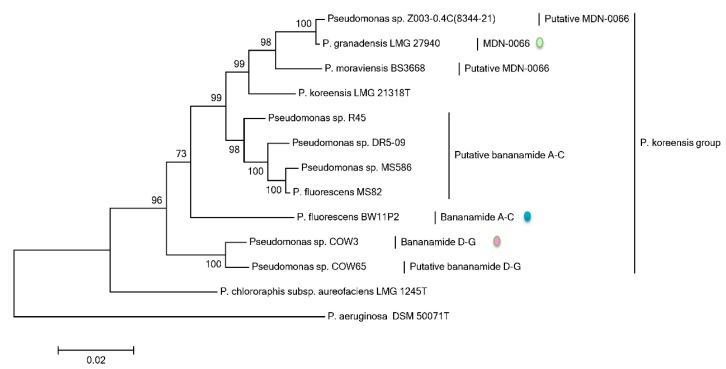
Multi-locus sequence phylogenetic analysis of *rpoD, rpoB,* 16S rDNA and *gyrB* partial sequences of bananamide-producing *Pseudomonas* strains. The tree was constructed with MEGA6 using the Maximum Likelihood Method with 1000 bootstrap replicates. Only bootstrap values above 70% are indicated. The CLPs with the coloured spheres have been fully characterized.

**Figure 7 molecules-24-04170-f007:**
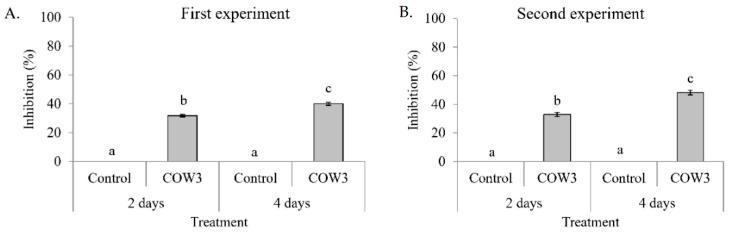
In vitro antagonistic effect of COW3 on *P. myriotylum*. Co-culturing of bananamide D-G-producing strains COW3, with *P. myriotylum* CMR1. (**A**) Percentage inhibition of CMR1 by COW3. Representative pictures were taken after two and four days, respectively. The experiment was conducted with 6 replicates and was repeated in time. The two experiments are shown. Vertical lines indicate standard deviations. Different letters (a, b, c) indicate statistically significant differences among different treatments (Anova followed by a Tukey’s tests: *n* = 6; α = 0.05), D: bananamide D; E: bananamide E; F: bananamide F; G: bananamide G. (**B**) Representative pictures showing the inhibition of *P. myriotylum* CMR1 by COW3.

**Figure 8 molecules-24-04170-f008:**
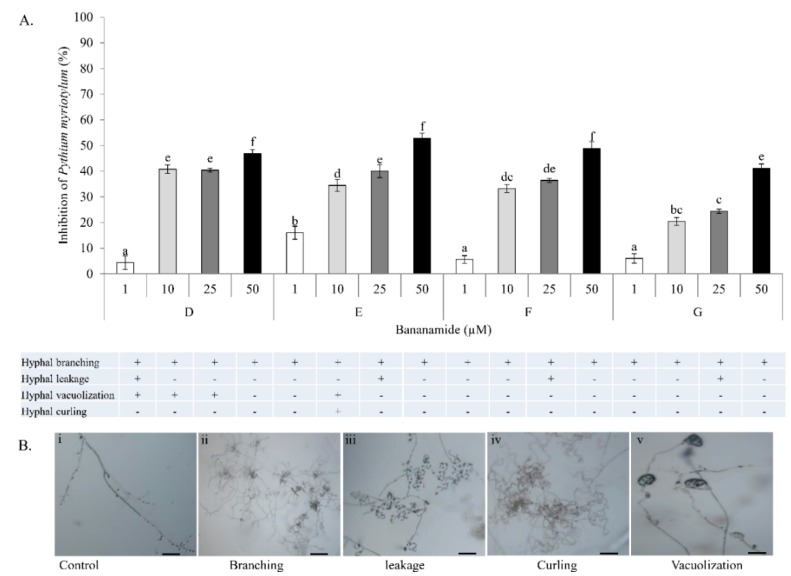
Effect of purified bananamide D-G on *P. myriotylum* CMR1. (**A**) Interaction between CLPs and *P. myriotylum* at concentrations ranging from 1 to 50 μM. Values plotted represent the percentage growth inhibition of *P. myriotylum* relative to the control. Vertical lines indicate standard deviations. Different letters (a, b, c, d, e, f) indicate statistically significant differences among different treatments (Anova followed by a Tukey’s tests: *n* = 5; α = 0.05), D: bananamide D; E: bananamide E; F: bananamide F; G: bananamide G. (**B**) Representative pictures showing various microscopic effects of pure bananamide D, E, F and G on *P. myriotylum* CMR1. Scale bars represent 10 μm.

**Figure 9 molecules-24-04170-f009:**
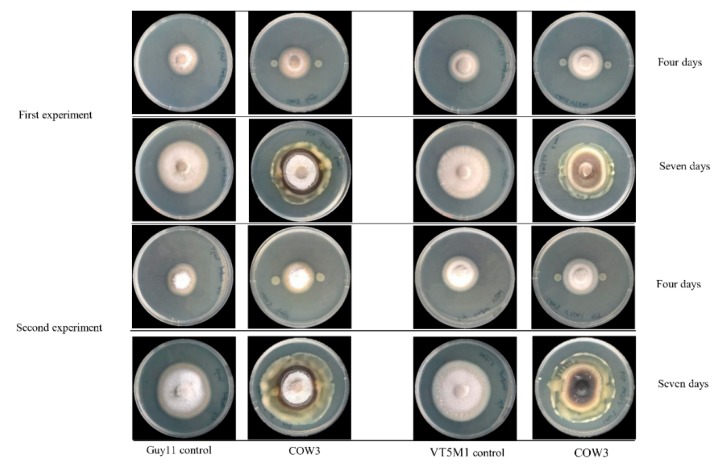
COW3 in vitro antagonistic test against blast fungi. Co-culturing of bananamide D-G-producing strain COW3, with *P. oryzae* VT5M1 and *P. oryzae* Guy11. Representative pictures were taken 4 and 7 days, respectively, after the pathogen was added to the center of the Petri dish. The experiment was carried out with 6 repetitions and was repeated in time.

**Figure 10 molecules-24-04170-f010:**
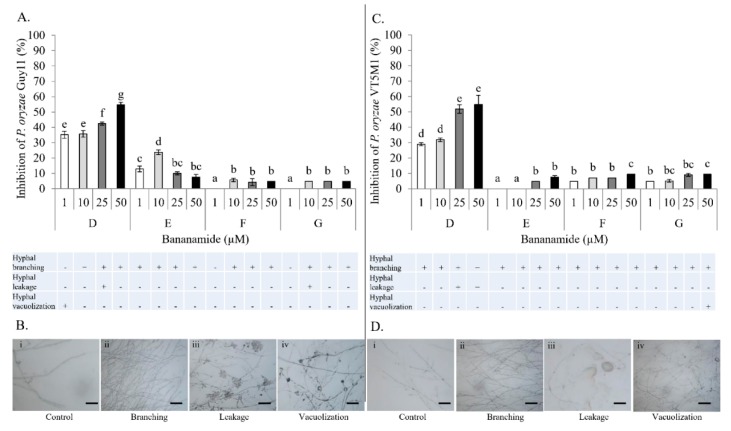
Effect of purified bananamide D-G on *P. oryzae* VT5M1 and Guy11. (**A**) Inhibitory interaction between CLPs and *P. oryzae* Guy11 at concentrations ranging from 1 to 50 μM. Values plotted represent the percentage growth inhibition of *P. oryzae* Guy11 relative to the control. Vertical lines indicate standard deviations. Different letters (a, b, c, d, e, f) indicate statistically significant differences among different treatments (Anova followed by a Tukey’s tests: *n* = 5; α = 0.05), D: bananamide D; E: bananamide E; F: bananamide F; G: bananamide G. (**B**) Representative pictures showing the various microscopic effects of pure bananamide D, E, F and G on *P. oryzae* Guy11. Scale bars represent 10 μm. (**C**) Interaction between CLPs and *P. oryzae* VT5M1 at concentrations ranging from 1 to 50 μM. Values plotted represent the percentage growth inhibition of *P. oryzae* VT5M1 relative to the control. Vertical lines indicate standard deviations. Different letters (a, b, c, d) indicate statistically significant differences among different treatments (Anova followed by a Tukey’s tests: *n* = 5; α = 0.05), D: bananamide D; E: bananamide E; F: bananamide F; G: bananamide G. (**D**) Representative pictures showing the various microscopic effects of pure bananamide D, E, F and G on *P. oryzae* VT5M1. Scale bars represent 10 µm.

**Table 1 molecules-24-04170-t001:** [M + H]^+^ values extracted from HRMS-spectrum and their corresponding molecular formula (MF) assigns using Kendrick Formula Predictor (https://bioinfo.cristal.univ-lille.fr/kendrick-webapp/).

m/z, [M + H]^+^	z	Monoisotopic Mass (m)	Theorical [M + H]^+^	Kendrick Putative MF	Name	Norine ID
1065.6862	1	1064.66072	1065.66800	C_53_H_92_N_8_O_14_	Bananamide D	-
1067.6967	1	1066.67637	1067.68365	C_53_H_94_N_8_O_14_	Bananamide E	-
1079.6995	1	1078.67637	1079.68365	C_54_H_94_N_8_O_14_	-	-
1081.7089	1	1080.69202	1081.69930	C_54_H_96_N_8_O_14_	-	-

**Table 2 molecules-24-04170-t002:** Amino acid sequences of bananamide D-G CLPs and related bananamides. The fatty acid is a 3-hydroxy fatty acid with either 10 or 12 carbons and either 0 or 1 unsaturation (double bond). Amino acids forming the macrocycle are underlined.

Strains	CLP *	*m*/*z*	Fatty Acid	1	2	3	4	5	6	7	8	References
*P. granadensis* LMG 27940	MDN-0066	-	3-OH C10:0	Leu	Glu	Thr	Leu	Leu	Ser	Leu	Ile	[39]
*Pseudomonas* sp. BW11P2	Bananamide A (1)	1108	3-OH C12:0	Leu	Asp	Thr	Leu	Leu	Gln	Leu	Ile	[34]
	Bananamide B (2)	1106	3-OH C10:0	Leu	Asp	Thr	Leu	Leu	Gln	Leu	Ile	[34]
	Bananamide C (3)	1080	3-OH C12:1	Leu	Asp	Thr	Leu	Leu	Gln	Leu	Ile	[34]
*Pseudomonas* sp. COW3	Bananamide D	1065.6	3-OH C12:1	Leu	Asp	Thr	Leu	Leu	Ser	Leu	Ile	This study
	Bananamide E	1067.6	3-OH C12:0	Leu	Asp	Thr	Leu	Leu	Ser	Leu	Ile	This study
	Bananamide F	1039.5	3-OH C10:0	Leu	Asp	Thr	Leu	Leu	Ser	Leu	Ile	This study
	Bananamide G	1051.5	3-OH C12:1	Leu	Asp	Thr	Leu	Leu	Ser	Leu	Val	This study

* Banamides A–C are called bananamide 1–3 in [34].

**Table 3 molecules-24-04170-t003:** Strains used in this study

Strain	Relevant Characteristics	Reference
*Pseudomonas sp.*		
COW3	Biocontrol strain isolated from the roots of cocoyam, N3 producer	[9]
COW65	Biocontrol strain isolated from the roots of cocoyam, N3 producer	[9]
*Pythium myriotylum*		
CMR1	Cocoyam pathogen causing root rot from Cameroon	[55]
*Pyricularia oryzae*		
VT5M1	Rice blast pathogen from Vietnam	[57]
Guy11	Rice blast pathogen from French Guyana	[58]

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
