# Peer review of "Pseudomonas sp. COW3 Produces New Bananamide-Type Cyclic Lipopeptides with Antimicrobial Activity against Pythium myriotylum and Pyricularia oryzae"

_molecules, 2019, doi:10.3390/molecules24224170_

Round 1

Reviewer 1 Report

This paper investigates the structure of several cyclic lipopeptides isolated from some bacteria using spectroscopic techniques (LC-MS, NMR). Computational techniques were used to determine where in the genome these lipopeptides are encoded. The activity of the lipopeptides against some other bacteria and fungi was investigated.

The details of the investigation are mainly standard. However, the authors claim in their conclusions that the "KMD approach [Chevalier et al. in press] can be employed for the discovery of novel CLP (lines 620-621). However, it is not obvious to this reviewer whether this is actually the case, without having access to this unpublished manuscript. In the present form of the paper, the statement in lines 620-621 is unconvincing and should be deleted. If the authors still want to include this statement, many more details need to be added.

Specific comments:

Lines 270+303: What does "with 6 repetitions and was repeated in time" mean? If the experiments were repeated more than 6 times, why not report all experiments?
Figures 7, 8, 10: Show error bars
Line 464: What are "sensitivity modes"? Are these different from positive or negative modes?
Line 470: Define the abbreviation "ACN" at the first occurrence.
Line 473: What are "NORINE-referenced compounds"? It is insufficient to refer to an unpublished report ("in press"). Uncommon procedures need to be explained in detail. Why was this procedure used and not some other procedure? Where does the value of "0.28" in the equation line 477 come from?
Lines 492-498: What is a "type VL ESI detector"? That needs to be explained better. Was this detector actually used? It says in line 498 that the signal was detected "using a diode array detector at a wavelength of 214 nm".
Line 508: Please specify the "Standard pulse sequences as present in the Bruker library" that were used.

Reviewer 2 Report

The manuscript presented by the authors describes four new CLPs named bananamides D-G, a gene cluster encoding the biosynthesis of these molecules and biological activities against P. myriotylum and M. oryzae. The manuscript is well presented and deserves publication in Molecules after minor revision.

The introduction should include structures for Bananamides A-C and MDN-066 as the new bananamides D-G are frequently referred to in the results and discussion.

What were the titres of the purified bananamides?

Can the authors comment on the identification of the other molecules presented in Fig 1 / Table 1?

Bananamide G differs from D-F by the inclusion of Val instead of Ile, and D/E/F differ with the fatty acid unit selected by the C(starter) domain . presumably the NRPS BanA-C synthesizes all versions? Are there any other NRPS candidates in the genome? And if not, can the authors explain why this NRPS is so flexible? Is this flexibility identifiable from the phylogenetic analysis of the A / C / whole NRPS? Including some details of the biosynthesis of other CLPs in the introduction may help frame this. 

One other point - Magnaporthe oryzae has recently been reclassified as Pyricularia oryzae. 

Reviewer 3 Report

This manuscript reports the discovery of bananamides type cyclic lipopeptides with antimicrobial activity from Pseudomonas sp. COW3. This study revealed complementarity of chemical analyses and genome mining in the discovery and elucidation of new peptides. Based on the structural novelty, logical approach for the discovery, and antimicrobial activity, the reviewer suggests that this manuscript can be a significant contribution to Molecules after minor revision.

When talking about the natural products structures, we try to use “new” instead of “novel”. There is no clear definition for what kind of skeleton can be described as “novel”. Please change the “novel” in the title to “new”. For most systems, the correct formulae cannot be determined solely on the basis of mass accuracy of monoisotopic MS signals. Although a number of initiatives have endeavored to devise more efficient and accurate approaches to determining elemental composition by mass spectrometry; so far only the isotopic fine structures acquired by FT-ICR MS has the ability to provide the unambiguous assignment of the molecular formulas (MFs). In this paper, the author mentioned using Kendrick mass defect (KMD) analysis for MF assignment. I think the author should add more details of this approach, and give readers more information about how to get the only one MF based on the monoisotopic mass by the Kendrick Formula Predictor. It is not necessary to mention ROESY for 1H-1H. Page 4, change “1H-1H ROESY” to “ROESY”. Page 4, line 143, change bananamide D-G to bananamides D-G. When talking about D- and L- configuration, the D and L should use small caps. Table 2 is too wilde.

Round 2

Reviewer 1 Report

The authors have addressed the points raised by this reviewer and made the appropriate corrections.